# Can Citizen Science Contribute to Avian Influenza Surveillance?

**DOI:** 10.3390/pathogens12091183

**Published:** 2023-09-21

**Authors:** Irene Saavedra, Julio Rabadán-González, David Aragonés, Jordi Figuerola

**Affiliations:** 1Consejo Superior de Investigaciones Científicas, Estación Biológica de Doñana, C/Américo Vespucio 26, E-41092 Sevilla, Spain; jordi@ebd.csic.es; 2Observation.org, C/Fresno 9, Almensilla, E-41111 Sevilla, Spain; julio@observation.org; 3Remote Sensing and GIS Laboratory (LAST-EBD), Consejo Superior de Investigaciones Cientificas, Estación Biológica de Doñana, C/Américo Vespucio 26, E-41092 Sevilla, Spain; daragones@ebd.csic.es; 4CIBER Epidemiology and Public Health (CIBERESP), E-28028 Madrid, Spain

**Keywords:** avian influenza virus, citizen science platforms, outbreak surveillance, early warning system

## Abstract

Global change is an important driver of the increase in emerging infectious diseases in recent decades. In parallel, interest in nature has increased, and different citizen science platforms have been developed to record wildlife observations from the general public. Some of these platforms also allow registering the observations of dead or sick birds. Here, we test the utility of live, sick and dead observations of birds recorded on the platform Observation.org for the early detection of highly pathogenic avian influenza virus (HPAIV) outbreaks in the wild in Belgium and The Netherlands. There were no significant differences in the morbidity/mortality rate through Observation.org one to four weeks in advance. However, the results show that the HPAIV outbreaks officially reported by the World Organisation for Animal Health (WOAH) overlapped in time with sudden increases in the records of sick and dead birds in the wild. In addition, in two of the five main HPAIV outbreaks recorded between 2016 and 2021, wild Anseriformes mortality increased one to two months before outbreak declaration. Although we cannot exclude that this increase was related to other causes such as other infectious diseases, we propose that Observation.org is a useful nature platform to complement animal health surveillance in wild birds. We propose possible approaches to improve the utility of the platform for pathogen surveillance in wildlife and discuss the potential for HPAIV outbreak detection systems based on citizen science to complement current surveillance programs of health authorities.

## 1. Introduction

Emerging infectious diseases are an important threat to human and animal health [1], and zoonotic diseases have important socioeconomic impacts, threatening the development of countries and first world economies [2]. The number of reported outbreaks of avian influenza viruses (AIVs) and other emerging infectious diseases has increased in recent decades in Europe [3]. Influenza A viruses of the family Orthomyxoviridae are the only orthomyxoviruses known to naturally affect birds [4]. Although spillover events into humans are rare, the deadliest flu pandemics in recent human history have involved viruses of avian origin [5]. AIVs are classified into two categories in relation to the ability of virus to produce disease and mortality in poultry: low pathogenic avian influenza viruses (LPAIVs) and highly pathogenic avian influenza viruses (HPAIVs). LPAIVs cause no signs of disease or slight disease. However, LPAIVs on wildlife can produce some negative impacts on animal life cycles and population dynamics (i.e., reducing their foraging and migratory performance [6]). HPAIVs cause severe disease and high mortality rates in poultry. However, ducks and other waterfowl can be infected without signs of disease. LPAIVs replicate mainly on mucosal surfaces of the respiratory and/or gastrointestinal tracts. In contrast, some HPAIVs can have a much broader tissue tropism and can replicate in many internal organs [7]. However, HPAIV strains have primarily adapted to bind and replicate in cells of the respiratory track, and this adaptation may favour zoonotic transmission [8]. Thus, AIVs mainly infect intestinal and respiratory cells and the viruses are excreted in high concentrations in the faeces of birds. Many avian reservoirs are associated with aquatic habitats because the virus can be transmitted by the faecal-oral route and remain viable in the water for days or even longer periods depending on pH, salinity and temperature [9,10]. Anseriformes (mainly ducks and geese) and Charadriiformes (mainly gulls and shorebirds) are the main reservoirs of AIVs [11,12,13]. 

There is no clear evidence of how HPAIVs are maintained from year to year, but there is evidence for long-term viral persistence in cold waters in the north and AIV persistence in duck populations during all seasons of the year [10,11,12,13]. However, prevalence tends to be high in juvenile birds in the north and to decrease throughout the season and during migration southwards [14]. For example, the seasonal peak of HPAIV prevalence is reported to occur in shorebird populations during northern migration and in wild ducks during the southern migration [15]. The proportion of sick and dead birds reported can depend on the concentration of birds in favourable stopover or wintering sites during migration periods [13]. During migration, many Anseriformes and Charadriiformes perform regular long-distance movements connecting many bird populations in time and space, and as a result, infected birds can carry and transmit the pathogens to new areas [13]. Consequently, AIV geneflow is higher within than between major waterbird flyways [16]. A recent analysis of HPAIV circulation among wild birds concluded that geese and swans drive HPAIV H5 transmission into ducks, gulls and other birds, while gulls were responsible for rapid long-distance movements of the virus [17].

In addition to affecting wildlife conservation and biodiversity, HPAIVs cause economic losses in poultry [18,19] and have been the origin of zoonotic viruses that have originated important epidemics in humans [20]. For these reasons, AIVs are the focus of national surveillance efforts across the world. In the European Union, the Council Directive 2005/94/EC requires the EU Member States to perform surveillance for avian influenza. From 2018, the European Food Safety Agency (EFSA) has the mandate to collate, validate and summarise the data collected by each member country. For example, in 2020, 24,290 poultry establishments and 31,382 wild birds were sampled [21]. The proportion of poultry and wild birds tested for AIVs also varies between countries. In 2021, 60% of the poultry establishments sampled in Europe were in The Netherlands (n = 5144), Italy (n = 5095) and Romania (n = 4363). The countries with the highest number of wild birds sampled were Germany (n = 15,165), Italy (n = 4005), Poland (n = 1426), The Netherlands (n = 1149) and Norway (n = 1148) [22]. However, in some countries, only a tiny percentage of carcasses are tested, thus limiting the capacity to detect AIV outbreaks in wild birds. 

Surveillance and early detection are necessary to analyse possible changes in host species and risk assessments, plan appropriate responses to reduce the impact on animal and human populations, increase awareness and identify areas that should be prioritized for surveillance [23]. For these, it is important to detect as soon as possible unusual numbers of sick or dead wild birds. Data on sick and dead birds in the public domain often come from established networks of collaborations. The increasing interest of the population in biodiversity conservation and citizen science has generated platforms that, taking advantage of new technologies, collect an impressive quantity of information on the observation of biodiversity, and especially of birds around the globe [24,25]. Here, we test the utility of the information collected by a general platform for biodiversity monitoring for the early detection of AIVs in Europe. The aim of this study was to investigate the capacity of nature monitoring platforms and propose possible approaches to increase their usefulness for pathogen surveillance. 

## 2. Materials and Methods

The World Organisation for Animal Health (WOAH) is a public interface to extract officially validated animal health data reported since 2005. We only used the dates of HPAIV outbreaks in wild birds because only these cases were reported in Netherlands and Belgium from 2016 to 2021 by WOAH. During the outbreak periods, there were five subtypes of HPAIV outbreaks: H5N1, H5N4, H5N5, H5N6 and H5N8. In addition, the reports of sick and dead birds were compiled from the Observation.org platform, one of the largest nature platforms where citizens record wildlife observations around the world since 2006, with many participants in Europe [26]. Over the years, there has been an increase in the number of recorded observations, reaching a total of 217.4 million through the Observation International family of portals, of which over 121 million correspond to birds. The number of observations of Anseriformes and Charadriiformes per year in Europe increased from 1,772,320 recorded observations in 2016 to 2,824,377 recorded observations in 2021. We only selected reports from January 2016 to December 2021 because the participation in Observation.org was much lower before this period.

In Observation.org, participants can register whether the observation belongs to a live, sick or dead individual. We focussed on the observations of sick and dead Anseriformes and Charadriiformes because they are the main AIV reservoirs in nature [11,12,13]. Furthermore, the number of participants at Observation.org varies between countries in Europe, and the highest number occurs in The Netherlands (57.82% of participants and 55.76% of observations) and Belgium (30.44% of participants and 29.38% of observations). None of the other European countries accumulated more than 3% of participants or 4% of observations (Figure 1A). Consequently, we focussed on the observations of sick and dead Anseriformes and Charadriiformes in Belgium and The Netherlands from 2016 to 2021 (see Figure 1B). During this period, 46,367,150 observations of Anseriformes and Charadriiformes were recorded by 854,057 observers in Belgium and The Netherlands, and 23,166 of the observations were of sick and dead birds (43.49% Anseriformes and 56.51% Charadriiformes). A total of 171 observations were discarded because they do not identify birds at least to order level.

During the period studied, we examined the changes in the proportion (expressed as 1/10,000) of sick and dead birds for each month (Figure 2) or week (Appendix A) in the periods when WOAH reports an outbreak. For each month and week of the study period, we calculated the proportion (expressed as 1/10,000) of sick and dead birds as the ratio between “the number of sick and dead birds” and “the number of observations reported”. 

We analysed whether there were significant differences in the proportion of sick and dead birds in relation to months. First, we analysed whether there were differences in the proportion of sick and dead birds in relation to the month with a generalised linear model (GLM) fitted with the binomial distribution and a logit link function. However, the binomial models had a severe overdispersion, indicating a bad fit of the models. The proportions of sick and dead birds did not follow a normal distribution and were Box–Cox-transformed to ensure normality. A linear regression model was used to test whether there were differences between months, years and countries in the Box–Cox-transformed proportion of sick and dead birds of Anseriformes and Charadriiformes. We included the interaction between month and year to examine whether the proportion of sick and dead birds of different months differed between years. The emmeans package was used to perform post hoc comparisons among groups using the estimated marginal means. We also calculated the variance explained by the three independent variables and the interaction to estimate the percentage of variation in the dependent variable they explain.

In addition, to determine whether there was a significant difference in the proportion of sick and dead birds per week between the outbreak and nonoutbreak periods from 2016 to 2021, we used Welch’s *t*-test because the variances were unequal, and we could not use a Student’s *t*-test to compare the means between the two independent groups. Moreover, a paired Student’s *t*-test was employed to assess if there was a significant difference in the proportion of sick and dead birds during the first week of the outbreaks compared to the preceding weeks (one-, two-, three- and four-week periods prior to the outbreaks were tested).

Data analyses were performed using the statistical program R 4.1.3 [27].

## 3. Results

Anseriformes and Charadriiformes were the orders with the majority of HPAIV-positives (70.12% of Anseriformes and 9.40% of Charadriiformes). Therefore, these orders are important indicators of avian influenza virus circulation, especially the order Anseriformes. HPAIV-positive species of Anseriformes were from family Anatidae (70.12%). The most often reported affected Anseriformes species were *Cygnus olor* (14.46%), *Anser anser* (14.46%), *Branta leucopsis* (11.33%) and *Anas penelope* (4.82%). The main families of HPAIV-positive Charadriiformes were Scolopacidae (2.65%) and Laridae (6.01%). The main Charadriiformes species affected was *Numenius arquata* (1.45%), and the six most affected Laridae species were of the Larus genus (5.05%). 

Regarding data compiled from Observation.org by month, the results show that 39.79% of the variability in the proportion of sick and dead birds of Anseriformes and Charadriiformes was determined by country, 24.17% by month, 20.37% by the interaction between month and year, and only 1.40% of the variability was determined by year. As a result, from 2016 to 2021, the proportion of sick and dead birds of Anseriformes and Charadriiformes did not differ in relation to year (*χ*^2^ = 1.39, d.f. = 5, *p* = 0.24). The highest proportion of sick and dead birds was recorded in The Netherlands (0.7827% vs. 0.2173% in Belgium, *χ*^2^ = 197.84, d.f. = 1, *p* < 0.001).Moreover, there were significant differences between months (*χ*^2^ = 10.92, d.f. = 11, *p* < 0.001). The proportion of sick and dead birds was lower in April, May and June (0.0003%, 0.0002% and 0.0002%, respectively) than in the rest of the months (*p* < 0.05). The summer and winter months were the months with the highest proportion of sick and dead birds (more than 0.0006%), with July, August and November having the highest proportions (0.0007%, 0.0007% and 0.0009%, respectively). Therefore, strong seasonal patterns biased towards the summer and winter months were observed from 2016 to 2021. The interaction between month and year was significant (*χ*^2^ = 1.84, d.f. = 55, *p* = 0.008) because the highest proportion of sick and dead birds did not occur in the same month each year (Figure 2).

During the period 2016 to 2021, fifteen HPAIV outbreaks were officially reported in The Netherlands and five in Belgium. However, many of these outbreaks overlapped in time and were grouped into five main periods: (1) from November 2016 to July 2017; (2) from December 2017 to April 2018; (3) from August 2018 to September 2018; (4) from October 2020 to July 2021; and (5) from August 2021 to December 2021 (Figure 2).

Figure 2 shows that the highest monthly proportion of sick and dead birds reported by the Observation.org platform overlapped in time with the HPAIV outbreak periods officially reported in Belgium and The Netherlands (Figure 2). There was an increase in the proportion of sick and dead birds in January–March 2016 and January–March 2019, but no HPAIV events were officially reported. These increased mortality events may be due to unnoticed avian flu circulation, other infectious diseases, poisoning or more probably cold spells and/or sea storms that are usually associated with increased avian mortality rates [28,29,30]. We tested the possible incidence of sea storms in these two exceptional mortality periods by comparing the number of Charadriiformes and Anseriformes (Appendix A in Appendix A). As expected, January–March 2016 and January–March 2019 mortality involved a higher percentage of Charadriiformes (73% of Charadriiformes and 27% of Anseriformes in January–March 2016 and 80% of Charadriiformes and 20% of Anseriformes in January–March 2019), especially species associated with marine environments (e.g., *Uria aalge*, *Chroicocephalus ridibundus* or *Larus argentatus*). Therefore, increases in the proportion of dead and sick birds reported to Observation.org overlap with officially declared HPAIV outbreaks and other well-known causes of bird mortality such as cold weather spells and sea storms. For example, in the beginning of January 2019, a severe storm in the southern North Sea probably contributed to mass mortality on the Dutch coast and the Wadden Islands [31]. In late January 2019, a deep cyclone and a major windstorm developed west of the British Isles and Ireland and moved across the region toward the North Sea and Central Europe [32]. Figure 2 also shows that the proportion of sick and dead birds tends to decrease in spring, especially in April–June 2017 and May–June 2021. 

In addition, Figure 2 shows that an increase in the proportion of sick and dead birds was detectable one month before declaration of the outbreaks in periods 3 and 4, especially in Anseriformes (see Appendix A). H5N8 subtype was predominant in outbreak periods 1 and 4, H5N6 subtype in outbreak periods 2 and 3 and H5N1 subtype in outbreak period 5. Thus, increases in the proportion of sick and dead birds one month before declaration of the outbreaks were not related to the subtypes involved in HPAIV outbreaks and their virulence.

Regarding data compiled from Observation.org by week, there were significant differences in the proportion of sick and dead birds per week between the outbreak and nonoutbreak periods (t = 2.89, df = 212.32, *p* = 0.004), being higher in outbreak periods (Mean ± SE = 0.0591% ± 0.0056%) than in nonoutbreak periods (Mean ± SE = 0.0403% ± 0.0034%). However, there were no significant differences in the means of the proportion of sick and dead birds between the first week of the outbreaks and one week in advance (t = 1.86, df = 4, *p* = 0.14), the first week of the outbreaks and two weeks in advance (t = 1.78, df = 4, *p* = 0.15), the first week of the outbreaks and three weeks in advance (t = 1.76, df = 4, *p* = 0.15) and the first week of the outbreaks and four weeks in advance (t = 0.88, df = 4, *p* = 0.43). The proportion of sick and dead birds during the first week of the HPAIV outbreak and one to four weeks in advance was similar. 

## 4. Discussion

Some specific applications have appeared for the monitoring of disease in wild birds. Probably the more successful experience comes from the project FeederWatch, initially designed to monitor the distribution and abundance of birds in winter [33], but that was the basis for the House Finch Disease Survey that allowed for the monitoring of the spread and impact of *Mycoplasma galliseptum* outbreak among House Finch *Carpodacus mexicanus* in North America [34,35]. In addition to the Observation.org platform used in our study, other nature platforms allow the recording of live, dead or sick birds. Stranding.nl is an initiative of Observation International to record beached animals in the Dutch coast. Bird Track, developed by the British Trust for Ornithology (BTO) (currently linked to eBird, supervised by the Cornell Lab of Ornithology), and Avian Check (supervised by Minister of Agriculture, Food the Marine of Ireland) are also good examples of bird observation platforms. 

In nature platforms, intrinsic (e.g., bird size) and extrinsic factors (e.g., geographical variation) affect reporting rates of sick and dead birds. Previous studies show that Anseriformes and Charadriiformes are the orders with the highest AIV rate [11,12]. In recent years, most of AIV-positives in wild birds in Europe were found in waterfowl game birds and breeding geese [22]. 

Previous studies suggest that online social network data can predict official data up to one week in advance, collecting data in almost real time at a rapid and less inexpensive rate [36,37]. In this study, we have shown that officially declared HPAIV outbreaks overlap with strong increases in the proportion of dead and sick birds reported to Observation.org. It is important to note that only HPAIV strains pathogenic to wild animals are expected to generate these increased mortalities, while the circulation of LPAIV strains may be more difficult to detect in the wild unless animals are specifically tested for such pathogens. Although there were no significant differences in the morbidity/mortality through Observation.org one to four weeks in advance, for two of the five main outbreaks monitored (periods 3 and 4), increased Anseriformes mortality was detectable one month before declaration of the outbreaks. This increase could be related to other causes such as other infectious diseases because they are not related to the subtypes of HPAIV outbreaks or their virulence. However, the results suggest that Observation.org can be a useful tool to complement current AIV surveillance programs by allowing the identification of where and when Anseriformes mortality anomalies occur. Nature platforms such as Observation.org may provide an automatic early warning system by implementing algorithms that rapidly identify mortality anomalies and consequently facilitate the detection, collection and laboratory analyses of samples for infectious disease outbreak detection. 

In addition, we identified a decrease in the proportion of sick and dead birds during the spring 2017 and 2021. This decrease may depend on the concentration of birds. For example, many bird species migrate during spring months, decreasing the concentration of birds and reducing the opportunity for the transmission of pathogens in specific areas. Moreover, the breeding behaviour changes the social interactions and spatial distribution, decreasing the number of contacts between infected birds and limiting virus transmission. In addition, not all increases in mortality recorded in Observation.org were associated with registered AIV outbreaks. We have shown that the percentage of Charadriiformes was higher than the percentage of Anseriformes during these increases. Although it is possible that some outbreaks remained unnoticed, we consider that some of these increases in mortality were related to other causes such as cold weather spells and sea storms because the number of AIV-positives reported by the European Food Safety Authority (EFSA) is higher in Anseriformes than in Charadriiformes wild birds [21,22,38,39], although AIV outbreaks may have important impacts on seabirds, particularly at breeding colonies [40]. Thus, the number of sick and dead Anseriformes birds could be a useful factor for early detection of AIV outbreaks. We could complement surveillance programs of national health authorities reporting the increases in the mortality of Anseriformes recorded in Observation.org. In addition, increases in the mortality could be communicated through colour-coded maps on official websites or warning text messages could be sent to the subscribing public and poultry farmers to prevent transmission to domestic birds.

In addition to identifying increases in the proportion of sick and dead wild birds, we propose other strategies for the early detection and control of infectious disease outbreaks. For example, allowing nature platforms to include standardized information on the cause of death when a dead individual is reported or on the symptoms when a sick individual is reported may allow nature platforms to refine outbreak detection capacity by refining algorithms to detect increased mortality. This is because dead individuals due to hunting, collision with buildings or other human infrastructures or predation are unrelated to infectious disease outbreaks. In the same way, sick individuals with visible fractures have lower possibilities of being due to AIVs or other infectious diseases. We should improve nature platform tools to allow rapid identification of the cause of death to detect outbreaks and transfer this information to authorities [41]. 

Geographic variation in the use of platforms also affects reporting rates. Observation.org probably reflects the higher number of ornithologists and naturalists using Observation.org in Belgium and The Netherlands compared to other European countries. Other platforms have stronger implantation in other countries such as Ornitho.ch in Switzerland and Italy. Thus, direct comparisons between European countries should be avoided, and unlike data from different nature platforms could be integrated. Although we have restricted our analyses to Belgium and The Netherlands, other platforms and the continuous increase in the use of this application may allow us to extend such analyses to other countries. The collection of data on sick and dead individuals widens the potential uses of data provided by citizens and needs to be incorporated at the interfaces used by different platforms.

## 5. Conclusions

In conclusion, our work suggests that nature platforms may be useful to confirm and detect AIV outbreaks in wild birds. Nature platforms may not replace traditional surveillance systems, but Observation.org could be combined with existing systems to improve and complement outbreak detection and prediction in wild birds, by allowing rapid detection of where and when mortality anomalies are occurring. Automatic algorithms may be implemented, and the collection of additional data to discard nondisease-related deaths may improve the predictive capacity of citizen science-derived platforms. Citizen science may be an effective tool to optimize resources in those areas where resources available for sample collection and analyses are more limited by allowing direct efforts in the areas with higher wildlife mortality. 

## Figures and Tables

**Figure 1 pathogens-12-01183-f001:**
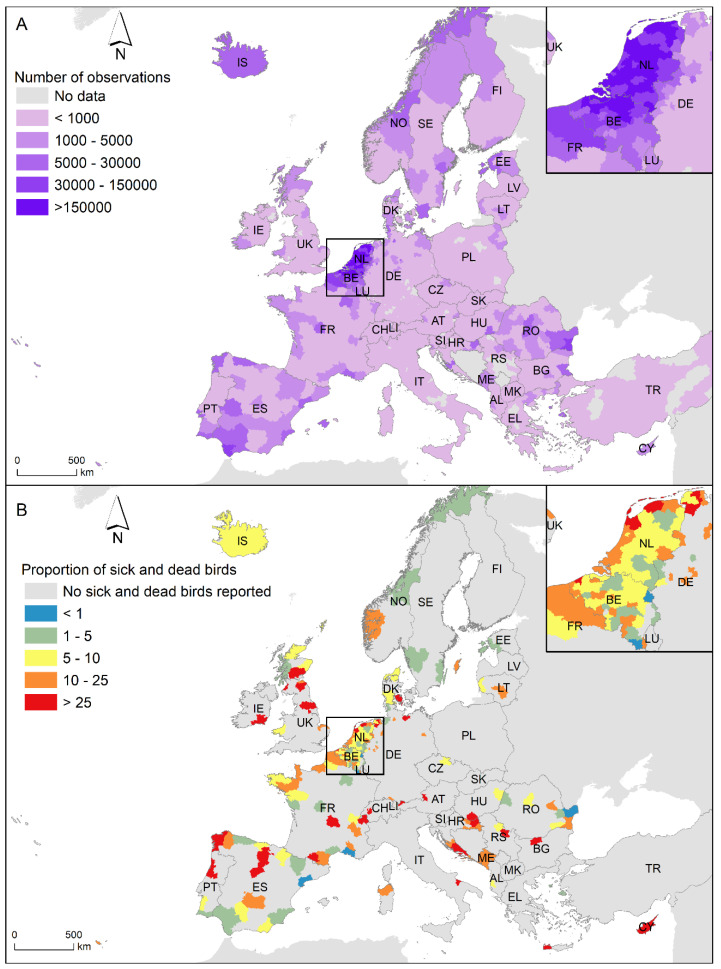
(**A**) Number of observations of Anseriformes and Charadriiformes by administrative unit (NUTS3 region) submitted to Observation.org between 2016 and 2021. The Netherlands and Belgium zoomed in on the upper left area. (**B**) Proportion of dead and sick Anseriformes and Charadriiformes per 10,000 observations by administrative unit (NUTS3 region) submitted to Observation.org between 2016 and 2021. The Netherlands and Belgium zoomed in on the upper left area.

**Figure 2 pathogens-12-01183-f002:**
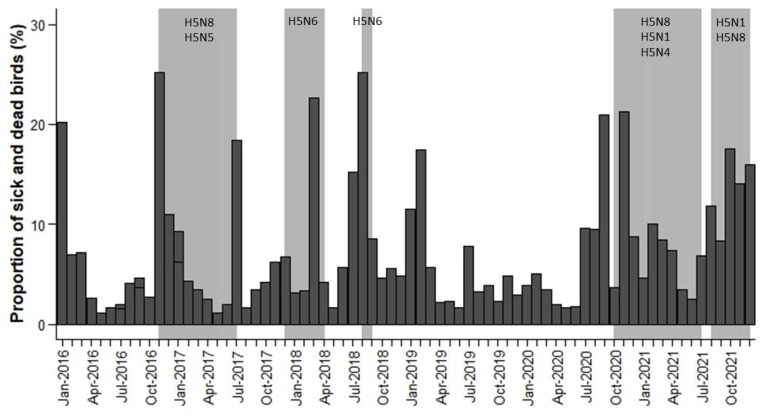
Temporal variation in the proportion of sick and dead birds (expressed as 1/10,000) by month (dark grey bars) and the periods of officially reported HPAIV outbreaks (light grey bars) and HPAIV subtypes.

## Data Availability

The original database is available through Observation.org (accessed on 8 August 2022), and the aggregated data used in the analyses are available through GBIF.

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
