# Peer review of "Can Citizen Science Contribute to Avian Influenza Surveillance?"

_pathogens, 2023, doi:10.3390/pathogens12091183_

Round 1

Reviewer 1 Report

The authors focused on the observations of sick and dead Anseriformes and Charadriiformes because they are the main AIV reservoirs in nature. For comparing with the reports from OIE-WAHIS, are they really good indicators for AIV? Which species were reported outbreaks most in OIE-WAHIS?

The authors agreed “increases in the proportion of dead and sick birds reported to Observation.org overlap with officially declared AIV outbreaks and other well-known causes of bird mortality such as cold weather spells and sea storms”. Also other diseases or infectious diseases might increase the number. So it hard to conclude that the increased Anseriformes mortality detected one month before declaration of the outbreaks was due to AIV. The increased mortality maybe a coincidence and need to be investigated.

Line 149, How many outbreaks according to OIE in Belgium and The Netherlands in that period?

Line 151-152, 91.59% is not all, remove it.

Line 166, Are the proportions mean of 2016-2021?

Figure 2, Add “%” to the Y-axis label.

Author Response

Response to Reviewer 1 Comments

Point 1: The authors focused on the observations of sick and dead Anseriformes and Charadriiformes because they are the main AIV reservoirs in nature. For comparing with the reports from OIE-WAHIS, are they really good indicators for AIV? Which species were reported outbreaks most in OIE-WAHIS?

Response 1: Anseriformes and Charadriiformes are good indicators for AIV because they are also the orders with the majority of AIV-positives in OIE-WAHIS. We have added in the Results that “According to OIE-WAHIS reports in Belgium and The Netherlands from 2016 to 2021, Anseriformes and Charadriiformes were the orders with the majority of AIV-positives (70.12% of Anseriformes and 9.40% of Charadriiformes). Therefore, these orders are important indicators of avian influenza virus, especially the order Anseriformes. AIV-positive species of Anseriformes were from family Anatidae (70.12%). The most affected Anseriformes species were Cygnus olor (14.46%), Anser anser (14.46%), Branta leucopsis (11.33%), and Anas penelope (4.82%). The main families of AIV-positive Charadriiformes were Scolopacidae (2.65%) and Laridae (6.01%). The main Charadriiformes species affected was Numenius arquata (1.45%) and the 6 most affected Laridae species were of the Larus genus (5.05%).”

Point 2: The authors agreed “increases in the proportion of dead and sick birds reported to Observation.org overlap with officially declared AIV outbreaks and other well-known causes of bird mortality such as cold weather spells and sea storms”. Also other diseases or infectious diseases might increase the number. So it hard to conclude that the increased Anseriformes mortality detected one month before declaration of the outbreaks was due to AIV. The increased mortality maybe a coincidence and need to be investigated.

Response 2: Thanks. We have clarified in the Discission that the increased Anseriformes mortality detected one month before declaration of the outbreaks could also be due to other infectious diseases, “Although there were no significant differences in the morbidity/mortality through Observation.org one to 4 weeks in advance, for two of the five main outbreaks monitored, increased Anseriformes mortality was detectable one month before declaration of the outbreaks. This increase could be related to other causes such as other infectious diseases. However, the results suggest that Observation.org can be a useful tool to complement current AIV surveillance programs by allowing identification of where and when Anseriformes mortality anomalies occur.”

Point 3: Line 149, How many outbreaks according to OIE in Belgium and The Netherlands in that period?

Response 3: We have added that “WOAH reported 15 AIV outbreaks in The Netherlands and 5 in Belgium between 2016 to 2021”.

Point 4: Line 151-152, 91.59% is not all, remove it.

Response 4: Thanks. We have removed it.

Point 5: Line 166, Are the proportions mean of 2016-2021?

Response 5: Yes, they are. We have explained that “Therefore, strong seasonal patterns biased towards the summer and winter months were observed from 2016 to 2021”.

Point 6: Figure 2, Add “%” to the Y-axis label.

Response 6: Thanks. We have added “%” to the figures and we have explained in the ms and in the legends that the proportion is expresed as 1/100.

Reviewer 2 Report

The manuscript entitled “Can citizen science contribute to avian influenza surveillance?” by Saavedra and colleagues is a well written study aiming at assessing the potential for nature platforms to help predict HPAIV outbreaks in wild birds. While the study is of obvious interest and well written, I have the following questions and suggestions:

1.      Major points:

-          The study would need to be clarified in its objectives and scope as I feel there is a difference between the Introduction focusing mainly on LPAIV of digestive tropism in wild birds and the rest of the article where clearly HPAIV cases are only looked at (and for which many are of respiratory tropism too or even mainly in some cases). The introduction section should be rewritten in view of the study really carried out (indeed with sick and dead birds notifications it is unlikely that LPAIV are detected).

-          I am surprised by the low number of reported outbreaks used for comparison with birds observations (20 in total over the 6 years period) while the EFSA report referenced in the introduction counts many more even just in 2021-2022 I believe (https://efsa.onlinelibrary.wiley.com/doi/epdf/10.2903/j.efsa.2022.7597). Would it be possible to use these numbers (quite precise too by month)? I feel it would make the comparison much more robust.

-          Figure 2: while the authors explain the 2 peaks of dead/sick birds observations without AIV outbreak reported with storms they do not explain low detection periods overlapping with AIV outbreaks (ex: spring 2017 or spring 2021), could you elaborate here?

-          Pathogenesis of HPAIV have differed in the different years (especially for Anseriformes). It would make sense to split the dataset here to take this important factor into account: one would expect precision of the prediction to be much better when HPAIV strains were very virulent in wild birds and less so when they were less virulent. (The pathogenesis data is available in the scientific literature for recent European strains.)

-          Discussion point: is there not a seasonal bias on when citizens tend to look at birds and report on nature platforms? (I would think vacation times (summer for example) would represent a peak of reporting?) Would this not impact the study outcome?

-          Is there a way to merge the different observational nature platform datasets? It would make the analysis again much more convincing.

-          Paragraph starting line 231: many examples can be found in human medicine as far as the benefits of looking at google searches (for example!) on epidemics onset (this is true and well-studied for seasonal flu for example). Did you look into incorporating this type of “citizen-based” data into the prediction here? Would hunters forums not be a good source of information too? Is it possible to aggregate these different general citizens data sources to make AIV outbreaks predictions better?

2.      Minor points:

-          Lines 36-38: virus and disease are mistaken for one another in this sentence I think.

-          Line 38: not all HPAIV cause high mortality in wild birds (especially in ducks): the definition of HPAIV is based on pathogenesis in chicken.

-          Line 47: the pH does also play a role in virus stability.

-          Line 52: reference 8 is correct but may not be the most up to date.

-       Sections numbering should be revised (all sections are numbered “1.”)

-          Please update OIE by WOAH throughout.

English typos could be corrected throughout (ex: “quantity of information” line 82; “pathogenS surveillance” line 87; “did” should replace “do” line 113;  “good examples FOR birds” line 225)

Author Response

Response to Reviewer 2 Comments

Point 1: The study would need to be clarified in its objectives and scope as I feel there is a difference between the Introduction focusing mainly on LPAIV of digestive tropism in wild birds and the rest of the article where clearly HPAIV cases are only looked at (and for which many are of respiratory tropism too or even mainly in some cases). The introduction section should be rewritten in view of the study really carried out (indeed with sick and dead birds notifications it is unlikely that LPAIV are detected).

Response 1: Many thanks, this suggestion has greatly improved our manuscript. We have rewritten and focused the Introduction and ms on HPAIV outbreaks.

Point 2: I am surprised by the low number of reported outbreaks used for comparison with birds observations (20 in total over the 6 years period) while the EFSA report referenced in the introduction counts many more even just in 2021-2022 I believe (https://efsa.onlinelibrary.wiley.com/doi/epdf/10.2903/j.efsa.2022.7597). Would it be possible to use these numbers (quite precise too by month)? I feel it would make the comparison much more robust.

Response 2: Most of the data collected by EFSA are reported by WOAH. EFSA recorded more outbreaks in 2021-2022 than our study because EFSA provided a broader overview of HPAI and LPAI outbreaks in poultry, captive and wild birds detected in Europe. However, we only focus on The Netherlans and Belgium from 2016 to 2021. During this period, 15 AIV outbreaks were officially reported in The Netherlands and 5 in Belgium. These outbreaks were grouped into 5 main periods because many of these outbreaks overlapped in time.

Point 3: Figure 2: while the authors explain the 2 peaks of dead/sick birds observations without AIV outbreak reported with storms they do not explain low detection periods overlapping with AIV outbreaks (ex: spring 2017 or spring 2021), could you elaborate here?

Response 3: We have added in the Results that “Figure 2 also shows that the dynamic of HPIAV outbreaks changed throughout the year. The proportion of sick and dead birds increased in winter and the proportion decreased in spring, especially in April-June 2017 and May-June 2021.”. Also, we have explained in the Discussion that “we identified a decrease in the proportion of sick and dead birds during the spring 2017 and 2021. This decreased may depend on the concentration of birds. For example, many bird species migrate during spring months, decreasing the concentration of birds and reducing the opportunity for the transmission of pathogens in specific areas. Also, the breeding behaviour changes the social interactions and spatial distribution, decreasing the number of contacts between infected birds and limiting virus transmission.”.

Point 4: Pathogenesis of HPAIV have differed in the different years (especially for Anseriformes). It would make sense to split the dataset here to take this important factor into account: one would expect precision of the prediction to be much better when HPAIV strains were very virulent in wild birds and less so when they were less virulent. (The pathogenesis data is available in the scientific literature for recent European strains.)

Response 4: Thanks, this suggestion has also greatly improved our manuscript. We have explained in the Material and Methods that “World Organisation for Animal Health (WOAH) is a public interface to extract officially validated animal health data reported since 2005. We only used the dates of HPAIV outbreaks in wild birds in Netherlands and Belgium from 2016 to 2021 because they were the only HPAIV reported in WOAH. During the outbreak periods, there were five subtypes of HPAIV outbreaks: H5N1, H5N4, H5N5, H5N6 and H5N8”. In addition, we have explained in the Results that “In addition, Figure 2 shows that an increase in the proportion of sick and dead birds was detectable one month before declaration of the outbreaks in periods 3 and 4, especially in Anseriformes (see Figure S2). H5N8 subtype was predominant in outbreak periods 1 and 4, H5N6 subtype in outbreak periods 2 and 3, and H5N1 subtype in outbreak period 5. Thus, increases in the proportion of sick and dead birds one month before declaration of the outbreaks are not related to the subtypes of HPAIV outbreaks and their virulence”. Also, we have added to the figures the labels of the outbreak subtypes present in each outbreak period. Finally, we have explained in the Discussion that “Although there were no significant differences in the morbidity/mortality through Observation.org one to 4 weeks in advance, for two of the five main outbreaks monitored (periods 3 and 4), increased Anseriformes mortality was detectable one month before declaration of the outbreaks. This increase could be related to other causes such as other infectious diseases because they are not related to the subtypes of HPAIV outbreaks and their virulence”.

Point 5: Discussion point: is there not a seasonal bias on when citizens tend to look at birds and report on nature platforms? (I would think vacation times (summer for example) would represent a peak of reporting?) Would this not impact the study outcome?

Response 5: We have explained in the Methods that “For each month and week of the study period, we calculated the proportion of sick and dead birds as the ratio between “the number of sick and dead birds” and “the number of observations reported”.”. The number of reported observations is closely related to the number of observers. Thus, in our opinion, the number of observations is a valid method to avoid the impact of seasonal reporting on natural platforms. However, previously we also used the "number of observers" to calculate the proportion of sick and dead birds and the results were very similar.

Point 6: Is there a way to merge the different observational nature platform datasets? It would make the analysis again much more convincing.

Response 6: We believe and hope that this article will promote the potential benefits of collaborations between different platforms to integrate and unify data.

Point 7: Paragraph starting line 231: many examples can be found in human medicine as far as the benefits of looking at google searches (for example!) on epidemics onset (this is true and well-studied for seasonal flu for example). Did you look into incorporating this type of “citizen-based” data into the prediction here? Would hunters forums not be a good source of information too? Is it possible to aggregate these different general citizens data sources to make AIV outbreaks predictions better?

Response 7: This is a preliminar study with one of the largest nature platforms where citizens record wildlife observations around the world but further studies are necessary to incorporate and agregate other citizien data sources to make outbreaks predictions better. We had added in the Discussion that “Direct comparisons between European countries should be avoided and unlike data from different nature platforms could be integrated. Although we have restricted our analyses to Belgium and The Netherlands, other platforms and the continuous increase in the use of this application may allow us to extend such analyses to other countries. The collection of data on sick and dead individuals widens the potential uses of data provided by citizens and needs to be incorporated at the interfaces used by the different platforms.”.

Point 8: Lines 36-38: virus and disease are mistaken for one another in this sentence I think.

Response 8: Many thanks. We have rewritten this paragrap (see in point 9).

Point 9: Line 38: not all HPAIV cause high mortality in wild birds (especially in ducks): the definition of HPAIV is based on pathogenesis in chicken.

Response 9: Thanks. We have rewritten this paragrap and we have included the definition of LPAIV  and HPAIV based on pathogenesis in chicken: "AIV are classified into two categories in relation to the ability of virus to produce disease and mortality in poultry: low pathogenic avian influenza viruses (LPAIV) and highly pathogenic avian influenza viruses (HPAIV). LPAIV cause no signs of disease or slight disease. However, LPAIV on wildlife can produce some negative impacts on animal life-cycles and population dynamics (i.e. reducing their foraging and migratory performance [6]). HPAIV cause severe disease and high mortality rates in poultry. However, ducks and other waterfowl can be infected without signs of disease. LPAIV mainly cause digestive tropism affecting the gastrointestinal tract of birds, and HPAIV mainly cause respiratory tropism affecting the respiratory tract and lungs.".

Point 10: Line 47: the pH does also play a role in virus stability.

Response 10: We have added that pH also play a role in virus stability and we have added two more appropriate references.

Point 11: Line 52: reference 8 is correct but may not be the most up to date.

Response 11: We have added references 9, 10 and 11.

Point 12: Sections numbering should be revised (all sections are numbered “1.”)

Response 12: Sorry, the sections were numbered correctly in the version of the manuscript we submitted. This error is only in in the PDF of the manuscript version created by the journal Pathogens.

Point 13: Please update OIE by WOAH throughout.

Response 13: Many thanks. We have changed OIE by WOAH throughout the manuscript.

Reviewer 3 Report

I read the ms with great interest. What I particularly like it the simplicity of the idea of using existing data sources to predict AIV outbreaks. Very useful for authorities and poultry farmers as they may keep their animals indoors before transmission to domestic birds occur. 

Perhaps a recommendation (if the authors have any ideas) about how best to communicate findings to farmers and public might be useful (a "colour code" communicated on a website or text messages sent to all subscribing farhers?)

Author Response

Response to Reviewer 3 Comments

Point 1: I read the ms with great interest. What I particularly like it the simplicity of the idea of using existing data sources to predict AIV outbreaks. Very useful for authorities and poultry farmers as they may keep their animals indoors before transmission to domestic birds occur.

Perhaps a recommendation (if the authors have any ideas) about how best to communicate findings to farmers and public might be useful (a "colour code" communicated on a website or text messages sent to all subscribing farhers?)

Response 1: Many thanks for your recommendation. We have added in the Discussion that “In addition, increases in the mortality could be communicated through colour-coded maps on official websites or warning text messages could be sent to the subscribing public and poultry farmers to prevent transmission to domestic birds.”.

Round 2

Reviewer 1 Report

No more comments.

Reviewer 2 Report

The authors took all comments and suggestions into account, the manuscript is improved.